# OpenReview forum: "UDora: A Unified Red Teaming Framework against LLM Agents by Dynamically Hijacking Their Own Reasoning"
_ICML.cc/2025/Conference — ICML 2025 poster_

### Official Review · Reviewer_Bj1V · 2025-03-10

**Overall Recommendation:** 1

**Summary:**

This paper proposes UDora, an iterative method based on GCG. The approach first collects responses z from the target victim model, then introduces a score function to modify the original response z into an attack-desired response z* (the response that achieves the attacker's goal). It then utilizes z* with the GCG algorithm to optimize the attack prefix. This process iterates until the model's output no longer requires modification and directly produces the attack-desired response z*. The authors conducted experiments on different datasets.

## Update after rebuttal

Dear authors,

The following is my further response to your rebuttal. I apologize that I didn't notice that you are not visible to the official comments.

******
Thank you to the authors for their efforts.

## C1: Unrealistic threat models
> We assume that an adversary can use the target LLM agent like any regular user.

I agree with this assumption. **However, this differs from what UDora assumes, which presumes that agents, environments, and user tasks are all accessible to attackers. UDora can directly access the agent responses, which is directly related to the environments (e.g., the tool call contents), user tasks, and the agent itself.**

In reality, attackers can interact with the target LLM agent, but they typically cannot access the actual environment the agent is interacting with. For example, in the case of a personal assistant agent, attackers can only attempt to simulate the user's computer environment. **The attackers can guess possible tool data or common user tasks, but cannot directly access the complete system.**

## C2: Difference with GCG
Thank you for the clarification. However, I still believe the main distinction between UDora and GCG is that UDora optimizes the suffix based on surrogate responses, which appears to be your primary motivation as well.

## C4: Regarding applying UDora to Agentdojo
The explanation that Agentdojo only supports API-based models seems insufficient to justify why UDora cannot be applied to it. Given that you successfully attacked GPT-4o on the WebShop dataset, it should be easy to conduct a small experiment applying UDora to Agentdojo using GPT-4o.

## Q1: Baselines
Regarding the false positive rates of GPT-4o and Claude 3.7, could you provide examples demonstrating why the injected prompts optimized by UDora are more stealthy? Additionally, I would appreciate seeing results using perplexity-based detection methods, as they have proven effective according to existing literature on jailbreaking.
******

Given these concerns, I maintain my original score.

**Claims And Evidence:**

Yes

**Essential References Not Discussed:**

The prompt injection attacks and defenses are largely not discussed.

**Experimental Designs Or Analyses:**

Yes. Please see the weaknesses part.

**Methods And Evaluation Criteria:**

Yes

**Other Comments Or Suggestions:**

The spacing appears compressed, for example in the Section 3 heading. I am unsure whether this violates submission requirements.

**Other Strengths And Weaknesses:**

# Strengths

The paper is clearly written.

# Weaknesses
## W1: Unrealistic threat model
The major scenario considered in this paper is essentially indirect prompt injection. In this field, attackers can only modify the environment—for example, as the authors mention, inserting certain text on a webpage—while having no knowledge of the user's agent (including the agent system architecture and backend LLM). However, UDORA requires knowledge of the user agent's background information and needs to access and modify the responses from the user's agent. If one can directly modify the user agent's response to obtain z*, why is it necessary to use GCG for iterative optimization? Why not simply replace z with z*?

## W2: Minimal differentiation from GCG
UDORA is fundamentally similar to GCG, with the only difference being that UDORA obtains the optimization target for GCG by accessing agent responses. Therefore, it is unsurprising that UDORA performs slightly better than GCG. For this same reason, UDORA requires gradient computation and thus cannot attack black-box models.

## W3: High computational complexity
The UDORA algorithm requires iterative sampling of z followed by GCG optimization of the adversarial suffix, resulting in extremely high computational complexity.

## W4: Insufficient experimentation
The evaluation only considers two 8B models, whereas in practical applications, agents typically use Claude or GPT-4, which UDORA cannot attack. Additionally, mainstream prompt injection datasets such as AgentDojo [1] were not evaluated. The dataset InjecAgent used in this paper contains only one step execution for every task, I am not sure if UDora can be applied to tasks that require multiple tool calls, such as AgentDojo.

[1] Debenedetti, Edoardo, et al. "Agentdojo: A dynamic environment to evaluate attacks and defenses for llm agents." NeurIPS 2024.

**Questions For Authors:**

Please refer to the weaknesses part.

**Relation To Broader Scientific Literature:**

This paper is related to prompt injection attacks and jailbreak attacks.

**Theoretical Claims:**

No theoretical claims are made in this paper.

---

> ### Author Rebuttal · Authors · 2025-04-01
>
> We thank the reviewer for recognizing the clarity of our writing and the relevance of our approach to prompt injection and jailbreak attacks. We appreciate your solid suggestions, which motivate us to further refine our work, and we are happy to address the concerns raised!
>
> > **Q1: Unrealistic threat model**
>
> Sorry for the confusion regarding the threat model. Different from GCG, for LLM agentic scenarios, we cannot simply optimize for an affirmative response to trigger a specific malicious action. Therefore, we propose leveraging the original reasoning of the LLM agent (for which we cannot directly modify it, but we can query it) based on the current adversarial string. We then craft a surrogate modified response, $z^*$, as the optimization objective—similar to the “Sure” objective in GCG. After training, we generate a response based on the optimized string to see if it indeed triggers the target malicious action. Thus, $z^*$ serves as an optimization objective similar to "Sure" in GCG, and we do not have direct access to modify the real LLM agent's response or replace $z$ with $z^*$.
>
> We hope this clarification resolves any confusion. If you have further questions, we are happy to provide additional details.
>
> > **Q2: Minimal differentiation from GCG**
>
> Thank you for raising this point about differentiating our work from GCG!
>
> GCG optimizes a fixed, affirmative prefix (e.g., "Sure"), which is static and suitable only for harmful requests. In contrast, UDora addresses the challenge that LLM agents perform extensive intermediate reasoning before executing their final action. Directly optimizing for the final action in this scenario with GCG is impractical. Instead, UDora uses a dynamic three-step optimization process:
>
> 1. **Gather the Initial Response**: We first query the initial response from the LLM.
>
> 2. **Constructing a Surrogate Optimization Target**: Leveraging the original response, we insert adversarial noise at optimal positions determined through a weighted interval algorithm guided by our positional scoring function (Equation 1). This modified surrogate response is hypothetical, not the actual response from the LLM.
>
> 3. **Optimizing the Adversarial String**: We then optimize the adversarial input string by maximizing our positional score rather than directly maximizing the probability as done in GCG. Note that, unlike GCG, we only leverage the gradient on the multiple noise insertions, rather than on the entire $z^*$.
>
> This adaptive and dynamic process enables UDora to exploit the LLM agent’s reasoning in a way that a static, fixed objective like GCG cannot. Furthermore, UDora also significantly outperforms GCG in empirical results. For instance, with the LLaMA-3.1 model, UDora achieves:
>
> - 99% ASR versus GCG’s 77% on the InjecAgent dataset,
> - 61.75% ASR versus GCG’s 15.00% on the WebShop dataset,
> - 97.7% ASR versus GCG’s 60.8% on the AgentHarm dataset.
>
> > **Q3: High computational complexity**
>
> We leverage KV caching for generating the $z$ in our implementation, and due to the word limit, we kindly encourage you to review the discussion about the computation cost provided in Q4 for Reviewer c28e.
>
> > **Q4: Insufficient experimentation**
>
> Thank you for this insightful suggestion! We indeed included a practical example (Fig 11) of attacking the AutoGen Email Agent from Microsoft, which employs GPT-4o as the base LLM (GPT-4 can also be applied), relying only on the returned log probabilities for the attack. Regarding models like Claude, where neither log probabilities nor tokenizers are available, the attack scenario becomes inherently more challenging and remains a direction for future exploration.
>
> Concerning the AgentDojo dataset, currently, it only supports API-based models, lacking direct compatibility with HuggingFace models, which doesn’t allow the attack optimization here. In addition, we also have communicated directly with the authors of AgentDojo, exploring possibilities for data extraction or integration with HuggingFace models. Unfortunately, both the authors from AgentDojo and we have agreed that certain technical difficulties (primarily related to prompt injections within pre-designed functions) have made this quite challenging. We appreciate your understanding of this limitation.
>
> Regarding the application of UDora to tasks involving multiple tool calls, our experiments with the AgentHarm dataset indeed encompass multi-step scenarios. We observed that once UDora successfully triggers the initial malicious tool call, subsequent steps often follow naturally. Thus, UDora can still effectively apply to multi-step tool call scenarios, as long as the first step is triggered. Moreover, adapting UDora explicitly for optimization across multiple steps is also feasible with minor code adjustments. Exploring and refining this multi-step optimization approach will be a focus of our next step, and your suggestion is greatly appreciated!

---

> > ### Comment · Reviewer_Bj1V · 2025-04-03
> >
> > Thanks the authors for the detailed rebuttal. While I appreciate the clarifications provided, I still have several concerns.
> >
> >
> > ## C1: Unrealistic threat models
> > First, in real-world applications, it is impractical to assume the attacker can access the agents' responses. In fact, attackers would never know when and where their injected instructions would be retrieved and processed by the agents. Reviewers VRpZ and c28e also acknowledge this unrealistic assumption.
> >
> > A more realistic threat model would be that attackers can only modify contents retrieved by tools (e.g., websites, emails, calendar events) without visibility into how the agent processes this information or what responses it generates. However, this is the most significant assumption that UDora made. Without access to agent responses, attackers cannot perform the optimization loop that UDora depends on.
> >
> >
> > ## C2: the difference between GCG and UDora
> > As stated by the authors, the only difference between UDora and GCG is that UDora optimizes the suffix based on surrogate responses. This is not a fundamental advancement.
> >
> > Also, it is not surprising that UDora outperforms GCG since UDora is optimized specifically on surrogate responses that are designed to mimic the targeted attack instructions.
> >
> >
> > ## C3: computational complexity
> > Thanks for providing the detailed computation costs. It confirms that for each injection and each user task, the attacker must search for a specific suffix to attack the agent. This requirement makes UDora impractical for real-world applications where attackers would need to generate customized attacks for countless potential user tasks without knowing which ones will be executed.
> >
> > ## C4: Regarding applying UDora to Agentdojo
> > I remain unconvinced about why UDora cannot be applied to AgentDojo. If there exists technical challenges preventing this application, it suggests that UDora cannot be easily deployed in real-world scenarios.
> >
> >
> > Also, I have a question regarding the defense:
> > ## Q1: A simple baseline: applying LLM to detect the tool retrieved contents.
> >
> > Regarding defense methods, thanks the authors for conducting experiments using LLaMA3.1 and Mistral models to detect potential prompt injections. I am curious whether more advanced models such as GPT-4o or Claude 3.7 might achieve better detection results with lower false positives. Additionally, have you explored perplexity-based detection methods?

---

> > > ### Author Response · Authors · 2025-04-03
> > >
> > > Thank you so much for your thoughtful suggestions. We truly appreciate the opportunity to discuss and clarify our setting and contributions with you!
> > > > **C1: Threat Model**
> > >
> > > Thank you for the follow-up questions! First, reviewers VRpZ and c28e raised the possibility of extending UDora to a black-box setting—where only the query (i.e., the agent’s response) is available—as a future research direction. **They specifically inquired about how our attack might be implemented with access only to the response, without access to corresponding gradients or log probabilities**. As discussed above and supported by our evaluation results, it is possible to extend our attack to these additional black-box scenarios. We are glad this point has been clarified, and we will incorporate this discussion into our revision.
> > >
> > > Additionally, we emphasize that access to the agent's response is both necessary and realistic in agent attacks, as noted by reviewers VRpZ and c28e. Specifically:
> > >
> > > 1. **Malicious Instruction Scenario:** In benchmarks such as AgentHarm, we assume the LLM agent’s user is malicious and intends to trigger harmful behavior. In such cases, it is natural and realistic for the adversary to access the agent’s responses like regular users. Under this threat model, we achieved a 97.7% attack success rate (ASR), compared to GCG’s 60.8%.
> > > 2. **Malicious Environment Scenario:** An adversary could access agent responses similarly to other users. For instance, APIs or SDKs for most LLM agents (e.g., OpenAI agent function calling) are publicly accessible. An adversary can perform routine tasks, gather responses, and subsequently optimize adversarial inputs based on these responses. Our evaluation against the real-world agent AI Email agents from AutoGen further validates the practicality of our attack.
> > >
> > > Moreover, our evaluation on black-box transferability-based attacks shown in Fig 10 (Perplexity AI) demonstrates that UDora successfully attacks realistic agent frameworks even under black-box conditions.
> > >
> > > We will add the suggested discussion to clarify our threat model in the final version.
> > >
> > > > **C2: Differences from GCG**
> > >
> > > Thanks for the question. As acknowledged by other reviewers, our work offers several key contributions beyond GCG:
> > >
> > > 1. **Adaptive Optimization:** We depart from GCG’s fixed-prefix optimization, which is challenging to adapt across different base LLMs and scenarios, requiring manually defined prefixes tailored to specific reasoning styles.
> > > 2. **Leveraging LLM’s Own Reasoning:** We propose a novel algorithm that exploits the LLM agent’s reasoning process to hijack itself. However, defining an appropriate optimization objective based on this reasoning presents a significant challenge, which we address comprehensively below.
> > > 3. **Optimal Multi-Position Noise Insertion:** To overcome this challenge, we introduce a systematic method for inserting noise at multiple positions within the reasoning process. Our algorithm optimally places $k$ instances of noise into the agent’s reasoning steps to create a robust surrogate optimization objective. Our experiments also demonstrate that both the number and the specific positions of inserted noises significantly affect attack effectiveness—contrary to the common assumption that a single targeted noise is sufficient.
> > > 4. **Targeted Gradient Utilization:** Unlike GCG, which uses gradients from the entire affirmative response for optimization, we exclusively utilize gradients from targeted noise and employ a positional scoring function to select optimal candidate strings. This modification significantly enhances the efficacy of adversarial attacks on LLM agents.
> > > 5. **Superior Performance:** We achieve considerably higher ASRs compared to GCG, surpassing it by at least 30% on average.
> > >
> > > All these contributions underscore UDora's substantial advancements over GCG.
> > >
> > > > **C3: computation cost**
> > >
> > > We appreciate the reviewer's acknowledgment of our detailed presentation on computational costs! For other experimental concerns, please refer to our response in C1.
> > >
> > > > **C4: Agentdojo**
> > >
> > > After consulting with AgentDojo's authors, the issue lies not in using UDora with AgentDojo, but in a compatibility limitation: AgentDojo currently only supports API-based models, not Huggingface models. This reflects an incompatibility between AgentDojo and Huggingface, not a limitation of UDora. Additionally, we have successfully applied UDora to three widely-adopted datasets—InjecAgent, WebShop, and AgentHarm—proving its effective real-world deployment.
> > >
> > > > **Q1: Baseline**
> > >
> > > Thanks for your insightful suggestion! Here are the results for GPT-4o and Claude 3.7 on the WebShop: true positive rates are 76.5% and 94.1%, respectively, but the false positive rates are still high at 61.5% and 80.8%. It is as challenging to detect hallucinations in an LLM agent. Perplexity-based methods will work, and our next step will be to refine our focus on more semantic strings. Any further suggestions would be appreciated!

---

### Official Review · Reviewer_vE1i · 2025-03-12

**Overall Recommendation:** 4

**Summary:**

The paper presents UDora, a unified red teaming framework designed to attack LLM agents by dynamically leveraging their reasoning processes. The core idea involves inserting adversarial perturbations into the agent's reasoning traces to steer it toward malicious actions. The framework operates in three steps: gathering initial responses, identifying optimal noise insertion positions, and iteratively optimizing adversarial strings. Experiments on three datasets (InjectAgent, WebShop, AgentHarm) demonstrate high attack success rates (ASR) across different scenarios (malicious environments/instructions) and LLMs (Llama 3.1, Ministral). UDora outperforms baselines like GCG and prompt injection, achieving up to 99% ASR. A real-world attack on an AutoGen email agent further validates its practicality.

**Claims And Evidence:**

The claims are well-supported by empirical evidence:

Effectiveness: Tables 1–3 show UDora’s superiority over baselines.

Generalization: Success across diverse scenarios (malicious environments/instructions) and models (open-source/closed-source) is demonstrated.

Real-World Applicability: The AutoGen email agent case study (Figure 11) provides concrete evidence of practical impact.

However, the theoretical justification for the positional scoring function (Equation 1) is underdeveloped. While empirically effective, its design choices (e.g., averaging token probabilities) lack formal analysis

**Essential References Not Discussed:**

Defensive Methods: Works like adversarial training (Madry et al., 2018) or detection mechanisms (Jones et al., 2023) are not discussed.

Multi-Agent Reasoning: Techniques from cooperative AI (e.g., debate frameworks) could contextualize UDora’s adversarial focus.

**Ethical Review Flag:**

Flag this paper for an ethics review.

**Experimental Designs Or Analyses:**

Strengths: Extensive experiments cover multiple models, datasets, and scenarios. Ablation studies (Tables 4–5) explore optimization modes and noise locations.

Weaknesses: Limited discussion of computational costs (e.g., iteration steps, token sampling overhead).

**Methods And Evaluation Criteria:**

Methods: UDora’s dynamic optimization strategy is novel, particularly its use of sequential/joint noise insertion to exploit reasoning paths. The integration of token-level probability distributions enhances adaptability.

Evaluation: ASR is a clear metric, but additional metrics (e.g., robustness to defenses, transferability) would strengthen the analysis.

**Other Comments Or Suggestions:**

Reproducibility:

While the appendices provide examples, releasing code or pseudocode for the optimization process (Algorithm 1) would enhance reproducibility.
Details on hyperparameter tuning are sparse.

Ethical Considerations:

The paper briefly acknowledges risks in the Impact Statement but lacks actionable mitigation strategies (e.g., controlled release of adversarial strings, collaboration with red- teaming communities).
 Including a discussion on potential defenses (e.g., adversarial detection, reasoning path validation) would balance the focus on attack efficacy.

Suggestions for Improvement:
1. Defense Discussion: Expand Section 7 to include preliminary experiments on defending against UDora-style attacks.
2. Code Release: Provide a minimal implementation or pseudocode for key components (e.g., positional scoring).
3. Failure Analysis: Include examples of unsuccessful attacks to identify UDora’s limitations (e.g., cases where reasoning paths resist perturbation).

**Other Strengths And Weaknesses:**

Strengths:

Novel framework addressing underexplored LLM agent vulnerabilities.
High practical impact with real-world case studies.

Weaknesses:

The paper lacks theoretical guarantees.
Limited exploration of ethical implications (e.g., misuse risks).
Overemphasis on ASR without analyzing failure modes or attack detectability.

**Questions For Authors:**

1. Theoretical Basis: Can you formalize the conditions under which UDora’s noise insertion guarantees successful attacks?
2. Defense Evaluation: How does UDora perform against LLM agents with adversarial training or detection mechanisms?
3. Ethical Safeguards: What measures are proposed to prevent misuse of UDora?

**Relation To Broader Scientific Literature:**

The work builds on prior adversarial attacks (GCG, AutoPrompt) and LLM agent benchmarks (WebShop, AgentHarm). It extends red teaming to dynamic reasoning exploitation, addressing gaps in fixed-prefix optimization.

**Theoretical Claims:**

The paper lacks theoretical guarantees. While the attack algorithm is empirically validated, formal analysis of how noise insertion impacts reasoning fidelity is missing.

---

> ### Author Rebuttal · Authors · 2025-04-01
>
> We sincerely thank the reviewer for recognizing UDora’s innovative approach in utilizing reasoning steps and for highlighting the robustness of our empirical results across diverse benchmarks, particularly in the real-world AutoGen email agent attack scenario. We deeply appreciate your thorough and thoughtful review, which has significantly strengthened the clarity and impact of our contributions!
> > **Q1: While positional scoring function is empirically effective, its design choices lack formal analysis**
>
> Thank you for highlighting this point; we will include a more formal analysis to clarify our design choices in the final revision. Briefly, our positional scoring function (Equation 1) is guided by two primary principles: (1) we prioritize positions where some tokens already match the target; (2) if multiple positions have the same number of matched tokens, we prefer the one with the highest average token probability rather than the product, as the product would undesirably bias selections based on token length. Equation 1 precisely reflects these principles. We will provide additional intuition and a formal rationale for this approach in our final version.
>
> > **Q2: Limited discussion of computational costs**
>
> Thank you for raising this important point! Due to the word limit, we kindly encourage you to review the statistic provided in Q4 for Reviewer c28e.
>
> > **Q3: Essential References Not Discussed**
>
> Thank you for suggesting these essential references for contextualizing UDora’s adversarial focus! We have added all these in our current version.
>
> > **Q4: Code Release: Provide a minimal implementation or pseudocode for key components (e.g., positional scoring).**
>
> Thank you for this insightful suggestion. We will release our code upon acceptance and also include the pseudocode for the positional scoring function, in the appendix of our final version.
>
> > **Q5: Failure Analysis: Include examples of unsuccessful attacks to identify UDora’s limitations**
>
> Thank you for the valuable suggestion! While our current analysis primarily focuses on successful attacks, we agree that examining unsuccessful cases is equally important for understanding UDora’s limitations. In line with our current detailed successful-case analyses (Fig 5–9), we will include representative failure examples from each dataset in our final version.
>
> > **Q6: Theoretical Basis: Can you formalize the conditions under which UDora’s noise insertion guarantees successful attacks?**
>
> Thank you for this insightful question! Empirically, we've observed that attacks typically succeed when the overall positional score across all noise insertion locations exceeds a threshold (e.g., greater than 3). Intuitively, this corresponds to cases where a specific target is mentioned multiple times during the LLM agent's reasoning, significantly increasing the likelihood that the agent adopts the associated malicious action. We hypothesize that this phenomenon arises because such repetitive mentions rarely occur in the LLM agent’s training data without the agent subsequently performing the mentioned action.
>
> > **Q7: Defense Evaluation**
>
> Thank you for the insightful question! Regarding adversarial training, it could mitigate UDora-style attacks, but it typically involves a trade-off, potentially reducing the overall utility or generalization capability of the LLM agent. For detection mechanisms, following your suggestion and reviewer c28e’s recommendation, we will expand Section 7 to include preliminary experiments on defense mechanisms against UDora-style attacks. Specifically, we plan to explore reasoning validation techniques, such as detecting hallucinated or erroneous steps introduced by UDora to achieve the target malicious action. For instance, we can prompt the LLM to self-assess the consistency between its reasoning steps and the original instruction or the observation, thereby evaluating its ability to recognize and mitigate UDora-style adversarial manipulations.
>
> We have included some initial results in our response to Q2 under Reviewer c28e. Due to the constraints of the word limit, we kindly invite you to examine the details there.
>
>
> > **Q9: Ethical Considerations and Safeguards**
>
> Thank you for raising this critical concern. We will carefully control the release of adversarial strings derived from real-world examples, such as those involving AutoGen demonstrated in our paper, to prevent potential misuse. Additionally, we will discuss potential mitigation strategies, including reasoning validation (as previously mentioned) and methods like obscuring or summarizing intermediate reasoning steps presented to users (like open o1) during interactions with LLM agents. Furthermore, we agree that collaboration with red-teaming communities and implementing adversarial detection mechanisms would significantly enhance security. We will include a new paragraph addressing these ethical considerations and proposed safeguards in our final revision.

---

### Official Review · Reviewer_c28e · 2025-03-13

**Overall Recommendation:** 3

**Summary:**

This paper introduces UDora, a novel red teaming framework designed to systematically attack LLM agents by leveraging their own reasoning processes. Unlike traditional adversarial attacks that rely on static prompt injections or optimized adversarial suffixes, UDora dynamically identifies and perturbs reasoning traces within LLM agents to optimize adversarial strings. The paper formulates the attack strategy as a multi-step iterative optimization process, showing superior attack success rates across multiple datasets (InjecAgent, WebShop, and AgentHarm). It further demonstrates real-world security implications by successfully misleading deployed AI agents like email assistants.

**Claims And Evidence:**

The claims in the paper are well-supported by experimental results. The evaluation on InjecAgent, WebShop, and AgentHarm benchmarks provides strong evidence for the framework’s effectiveness. However, while the attack success rates are high, it would be useful to include more ablations analyzing the robustness of different attack configurations (e.g., varying the number of noise insertion points in different attack settings).

**Essential References Not Discussed:**

There is no other essential references not disscussed.

**Experimental Designs Or Analyses:**

The experiments are generally well-designed, covering multiple attack scenarios. The dataset selection is appropriate, and the baselines are reasonable. However, it would be beneficial to explore how different LLM architectures (beyond Llama and Mistral) respond to UDora.

**Methods And Evaluation Criteria:**

The methodology is well-motivated, and the proposed noise insertion and adversarial string optimization strategies are well-explained. However, additional comparisons with alternative red teaming approaches (e.g., self-reflection-based attacks) would provide a broader context.

**Other Comments Or Suggestions:**

I have no  other comments or suggestions.

**Other Strengths And Weaknesses:**

The paper introduces a new framework that leverages LLM agents' own reasoning processes for red-teaming, which is a creative combination of adversarial attack techniques and reasoning-based optimization strategies.

However, a notable weakness is the lack of comparative analysis with other red-teaming frameworks specifically designed for LLM agents, which limits the contextual understanding of UDora's performance advantages. Additionally, the proposed method relies on access to token-level probability distributions, which may not always be available in black-box settings. The feasibility of attacking models that do not expose these probabilities remains unclear. The iterative attack strategy, while effective, could introduce significant computational overhead. While the authors demonstrate average number of optimization iterations in Figure3, it would be useful to quantify the trade-off between attack efficiency and computational cost more explicitly, for example, analyze the exact average running time.

**Questions For Authors:**

1. How would UDora perform in a black-box setting where token probabilities are not available?
2. Have you considered countermeasures such as reasoning validation to detect adversarial manipulations?
3. How does UDora compare against self-reflective adversarial attacks that use an LLM’s own safety filters to generate bypassing strategies?
4. What are the computational requirements for running UDora at scale? Could it be optimized for faster attack execution?

**Relation To Broader Scientific Literature:**

The paper builds upon adversarial attack techniques in NLP and extends them into the LLM agent domain. While it cites relevant prior work, additional discussion on adversarial reasoning attacks (e.g., those targeting CoT reasoning or multi-step planning in LLMs) could strengthen the contextualization.

**Theoretical Claims:**

The paper does not rely on formal theoretical claims, but the optimization-based attack procedure is sound. If possible, providing a theoretical justification for why adversarial noise placement at specific reasoning points is more effective than simple prompt injections would be valuable.

---

> ### Author Rebuttal · Authors · 2025-04-01
>
> We are deeply grateful to the reviewer for their thorough and insightful feedback. Your contribution of time and expertise has significantly enriched the development of our research!
>
> > **Q1: Black-box setting**
>
> Thank you for the insightful question! In a pure black-box environment—where only the final response is observable—the attack indeed becomes more challenging. Still, UDora could be extended in a few ways:
>
> 1. Surrogate Model Training: Train a surrogate LLM to mimic the target LLM agent’s behavior based on observed responses, then apply UDora on the surrogate LLM. The resulting adversarial prompt may transfer to the real target model.
>
> 2. Cross-Model Transferability: Leverage the fact that adversarial prompts sometimes transfer between models, as illustrated by our Perplexity AI example using open-source models (e.g., LLaMA 3.1 or even the currently widely adopted DeepSeek-R1).
>
> Extending UDora to a purely query-based, black-box attack is a natural and important next step, and we are actively exploring methods in that direction. We welcome any additional ideas you might have on this front!
>
> > **Q2: Reasoning validation to detect adversarial manipulations**
>
> Thank you for raising the idea of using reasoning validation as a countermeasure! Our experiments (Figs. 5–9) show that UDora’s manipulations distort the agent’s reasoning—effectively causing "hallucinated/ wrong" reasoning paths that steer the agent toward harmful actions. For instance, in the WebShop scenario, the agent is tricked into believing an incorrect item meets all specified requirements (e.g., price, attributes), resulting in a malicious action.
>
> Therefore, we conducted the following experiments on reasoning validation as follows: we used a separate LLM to independently determine if the reasoning from the LLM agent aligns with the instructions or the observation from the environment. Specifically, we found the detection rates to be 97.0% and 75.0% on the WebShop dataset using the LLaMA3.1 and Ministral models, respectively. However, we also observed a high false positive rate, as 69.2% and 89.2% of the cases with normal reasoning were also incorrectly flagged as incorrect reasoning, indicating that reasoning validation in practice is still somewhat challenging as it is as difficult as detecting hallucinations in LLM agents.
>
> > **Q3: Compare against self-reflective adversarial attacks**
>
> Self-reflective adversarial attacks focus primarily on bypassing an LLM’s safety filters to extract harmful responses. By contrast, UDora targets a malicious tool or functionality rather than just overcoming safety barriers.
>
> 1. **Malicious Environment:** In contexts like WebShop or InjecAgent, UDora aims to manipulate the LLM into taking a malicious action (e.g., buying something undesirable) without producing obviously harmful or unethical text. Because the output may look innocuous, standard safety filters that rely on detecting overtly harmful or unethical content often fail to catch this subtle manipulation.
>
> 2. **Malicious Instruction:** Here, self-reflective attacks stop at bypassing the filter, but UDora must also ensure the malicious function is genuinely activated. As shown in Fig. 8 (4), even if the request bypasses safety checks, it might not trigger the intended malicious action, requiring further optimization of UDora.
>
> Hence, UDora is more specialized than self-reflective methods: it not only defeats safety filters but also reliably orchestrates a specific malicious action.
>
> > **Q4: Computational requirements and faster attack execution**
>
> In UDora, the computational overhead at each iteration primarily stems from two parts:
>
> 1. **Query time:** Generating the model’s reasoning process given the current adversarial string.
> 2. **Optimization time:** Finding the optimal position, computing gradients, and updating the adversarial string.
>
> When running UDora with LLaMA 3.1:
>
> - On the AgentHarm dataset, the average times per iteration for these two parts are **3.48s** (query) and **11.22s** (optimization).
> - On the InjecAgent dataset, they are **5.57s** (query) and **5.97s** (optimization).
>
> With the Ministral model:
>
> - On AgentHarm, the corresponding times per iteration are **2.49s** (query) and **6.04s** (optimization).
> - On InjecAgent, they are **6.86s** (query) and **8.18s** (optimization).
>
> These results show that, similar to GCG, **the main bottleneck is in the optimization phase**. However, UDora typically only requires around 20 iterations for a successful attack (see Fig. 3).
>
> UDora can also be optimized for faster attack execution:
>
> 1. **Less frequent updates:** Instead of updating the reasoning process after every iteration, update it at a specified interval (e.g., every 10 steps).
>
> 2. **Partial reasoning generation:** Generating only the first 100 tokens of reasoning during query can still yield high attack success rates. For instance, on AgentHarm with LLaMA 3.1, generating just the first 100 tokens still achieves a 97% ASR.

---

### Official Review · Reviewer_VRpZ · 2025-03-14

**Overall Recommendation:** 3

**Summary:**

This paper introduces UDora, a unified framework for testing security vulnerabilities in LLM agents. It focuses on two scenarios: malicious environments and malicious instructions. UDora works by analyzing an agent's reasoning process, identifying optimal positions to insert misleading information, and optimizing attacking text through multiple iterations. UDora significantly outperformed existing methods across three datasets.

**Claims And Evidence:**

The claims made in this submission are supported by the experiments. This paper conducted experiments on three datasets (InjecAgent, WebShop, and AgentHarm) with different LLMs, demonstrating that UDora's attack success rate (ASR) is significantly higher than GCG and Prompt Injection attack methods.

**Essential References Not Discussed:**

N/A

**Experimental Designs Or Analyses:**

The experimental design is in general sound.

**Methods And Evaluation Criteria:**

The methods and the evaluation criteria (ASR) make sense.

**Other Comments Or Suggestions:**

N/A

**Other Strengths And Weaknesses:**

- UDora assumes the attacker can access either the entire model or its token probability distribution during reasoning, which is a strong assumption for real-world deployment.
- UDora lacks discussions on defenses.

**Questions For Authors:**

Would the ASB benchmark provide a more comprehensive evaluation framework for assessing the effectiveness of UDora?


*Zhang, Hanrong, et al. "Agent security bench (asb): Formalizing and benchmarking attacks and defenses in llm-based agents." arXiv preprint arXiv:2410.02644 (2024)*

**Relation To Broader Scientific Literature:**

UDora extends prior research on prompt injection by dynamically leveraging LLM agents' reasoning.

**Theoretical Claims:**

N/A

---

> ### Author Rebuttal · Authors · 2025-04-01
>
> Many thanks to the reviewer for the thoughtful and detailed feedback. The expertise and time invested in this work have been instrumental in enhancing its quality!
>
> > **Q1: UDora assumes the attacker can access either the entire model or its token probability distribution during reasoning, which is a strong assumption for real-world deployment.**
>
> We appreciate your insightful question regarding our assumption that attackers can access either the entire model or at least the token probability distribution! As discussed in the paper, this requirement is indeed a limitation in certain real-world settings—particularly for closed-source models (e.g., Claude) that do not expose log probabilities or even a tokenizer. However, we note that some open models (including GPT-series models) do provide token-level probabilities, which enables our proposed attack in practice. For instance, in our experiments, the GPT-4o-based AutoGen email agent made these probabilities accessible, facilitating UDora’s successful attack strategies.
>
> Meanwhile, we observe a broader trend of rapidly improving open-source LLMs (e.g., DeepSeek-R1) being integrated into real-world applications, even in partnership with major industry players (e.g., [Perplexity AI](https://www.perplexity.ai/page/deepseek-s-new-open-source-ai-YwAwjp_IQKiAJ2l1qFhN9g?login-source=oneTapPage&login-new=false), [NVIDIA](https://build.nvidia.com/deepseek-ai/deepseek-r1), [Microsoft](https://azure.microsoft.com/en-us/blog/deepseek-r1-is-now-available-on-azure-ai-foundry-and-github/), etc.) to reduce costs. As this trend continues, full or partial access to model internals (including token probabilities) may become increasingly common in LLM agentic systems. In such cases, UDora’s approach remains directly applicable. Extending UDora to a purely query-based attack is a natural and important direction as our future work, and we are also excited to explore this in our next step!
>
> > **Q2: UDora lacks discussions on defenses.**
>
> Thank you for raising this important point about defenses! We will add a new paragraph in our final version to discuss how to mitigate potential misuse of UDora. One straightforward and practical strategy is for the agent provider to share only a condensed or sanitized summary of the reasoning steps like openai o1 reasoning model—rather than the full chain-of-thought. By limiting visibility into the underlying reasoning process, attackers lose critical information for inserting malicious noise in the reasoning steps. In our final version, we will also discuss concurrent work on guardrails for LLM agents, such as [1, 2].
>
> Besides, following the suggestions from Reviewers c28e and vE1i, we have also explored defenses related to reasoning validation, i.e., using a separate LLM to determine if the attacked reasoning aligns with the instructions or environment. Specifically, we found the detection rates to be 97.0% and 75.0% on the WebShop dataset using the LLaMA3.1 and Ministral models, respectively. However, we also observed a high false positive rate, as 69.2% and 89.2% of the cases with normal reasoning were also incorrectly flagged as incorrect reasoning, indicating that reasoning validation in practice is still somewhat challenging as it is as difficult as detecting hallucinations in LLM agents.
>
> [1] Xiang, Z., Zheng, L., Li, Y., Hong, J., Li, Q., Xie, H., ... & Li, B. (2024). GuardAgent: Safeguard LLM agents by a guard agent via knowledge-enabled reasoning. arXiv preprint arXiv:2406.09187.
>
> [2] Luo, W., Dai, S., Liu, X., Banerjee, S., Sun, H., Chen, M., & Xiao, C. (2025). AGrail: A Lifelong Agent Guardrail with Effective and Adaptive Safety Detection. arXiv preprint arXiv:2502.11448.
>
> > **Q3: Would the ASB benchmark provide a more comprehensive evaluation framework for assessing the effectiveness of UDora?**
>
> Thank you very much for suggesting this reference! We agree that the ASB benchmark would indeed provide a comprehensive evaluation framework for UDora, and we plan to include experiments using it in our final version. Due to the limited rebuttal period, fully familiarizing ourselves with the framework’s code and integrating our attack will require some additional time.
>
> In the current submission, we have evaluated UDora across three diverse datasets, including InjecAgent, whose data characteristics look a bit similar to those of the ASB benchmark. Thus, the performance results obtained on InjecAgent could serve as a preliminary indicator of the expected performance on the ASB benchmark.
>
> If you have any further questions or suggestions, please feel free to let us know. Your feedback is greatly appreciated and will certainly help us improve our work a lot!

---

### Decision · Program_Chairs · 2025-05-01

**Decision:**

Accept (poster)

**Comment:**

This paper received mixed reviews.

First, there were some concerns about what exactly the threat model entails. I read the paper to understand this better. The authors must significantly improve their presentation of the threat model and clarity of their attack. I believe their attack model assumes access to the logits of an LLM, but does not allow modification of the actual responses from tools / LLM itself - this was the main source of confusion for the reviewer who recommended rejection. However, their confusion is indeed well-founded because the approach talks about "injecting noise' but it's not clear that they do not directly inject noise but rather find a prompt that can result in the noise injection.

The second concern raised was about better contextualization in the literature, discussing assumptions and limits of various other attacks etc. I agree that the paper can be significantly improved along this axis. Furthermore, the novelty of the actual method is somewhat limited, so such careful analyses of setups and careful explanation is all the more important. There are several other sources of improvement pointed out by reviewers and i recommend the authors to improve their presentation to incorporate this feedback.

However, their approach still presents an interesting paradigm of manipulating the "reasoning" process of LLMs and how that affects agents. The empirical results over a variety of domains seem generally compelling. As a result, I am recommending weak accept.